# Monitoring, Diffusion and Source Speculation Model of Urban Soil Pollution

**Zhichao Li [1], Wanchun Lu [1,\*] and Jilin Huang [2]**

[1]   School of Political Science and Public Administration, East China University of Political Science and Law, Shanghai 201620, China; lizhichao@ecupl.edu.cn
[2]   College of Environmental Sciences and Engineering, Peking University, Beijing 100871, China; jilinhuang@pku.edu.cn
\*   Correspondence: wanchunlu@pku.edu.cn; Tel.: +86-150-2185-8319

**Abstract:** The rapid industrialization of cities has brought many challenges to the environment and resources. Industrial wastes, automobile exhaust, coal combustion soot and other pollutants accumulate in urban soil, and the characteristics of urban soil are changed, causing many pollutants to accumulate in the urban soil environment. Heavy metals are toxic and harmful pollutants existing in soil that cannot be biodegraded or thermally degraded; thus, heavy metals pose a threat to environmental quality and humans. To solve the environmental pollution of soil heavy metals, we utilize kriging interpolation to determine the geological distribution of the environmental pollution of metal elements and analyze the main causes of soil heavy metal pollution. Next, the propagation characteristics and diffusion process of heavy metal pollutants are thoroughly analyzed; in addition, an improved one-dimensional convective dispersion model and an improved air subsidence model are established, and real urban soil data are taken as an example for the fitting test. The results show that the improved models that consider more factors, such as adsorption or decomposition factors during the process of convective dispersion, absorption and expulsion factors of the crop root and topographic factors and height changes during the process of air subsidence, are effective. This paper is helpful for distinguishing the primary pollution sources and migration routes of soil metal element pollution and provides a certain reference value for protecting the environment and reducing heavy metal pollution.

**Keywords:** pollution of soil heavy metals; kriging interpolation; convective dispersion model; air subsidence model

## 1. Introduction

A city is a highly civilized form of habitation created by human beings. The rapid development of the economy and society in addition to developments in industrialization have accelerated the population concentration in cities [1]. Simultaneously, the development of a city also brings many pressures and challenges to the resources and environment. During the process of rapid industrialization and urbanization, the urban environmental quality is affected by mankind with each passing day, and the urban ecology [2] has increasingly deteriorated. Industrial wastes, automobile exhaust, coal combustion soot and other pollutants accumulate in urban soil, and the characteristics of urban soil are changed, causing many pollutants to accumulate in the urban soil environment [3].

There are many kinds of soil pollutants, when the contents of harmful substances in urban soil exceeds its purge ability, the soil system will change, thereby increasing the cost of environmental degradation [4]. Among them, the content of heavy metal elements in the soil has undergone serious changes due to human activities (industry, household waste and agriculture). Some studies have

determined that anthropogenic activities are the main causes of metal pollution [5]. Spatial analysis shows that transportation [6], metal smelting [7], and industrial production [8] all cause the accumulation of heavy metal elements in the soil. These pollutants will continue to migrate and decay in the soil layer. After a period of time, if no measures are taken to prevent this phenomenon, they may contaminate the groundwater system [9]. In a landfill in Thessaloniki, northwest Greece, heavy metal contamination in the soil has exceeded the standard, which indicates that heavy metal ions will continue to diffuse and migrate into the soil, expand in the affected areas, endanger the land function, reduce the land use value, and even affect human health [10]. Therefore, tracing the source of heavy metal pollution, cutting off or preventing pollution transmission in a timely manner, minimizing the possibility of harm, determining the main pollution source and its migration path, and controlling the location of the pollution source has become the key to controlling the soil environment.

At present, there is some accumulation of methodology for analyzing ground metal element pollution. The geoaccumulation index (Igeo), pollution index (PI), and potential ecological risk index (PER) were calculated to assess the pollution level in soils. The hazard index (HI) and carcinogenic risk (RI) were used to assess the human health risk of heavy metals [11]. The migration of metals in the ground is a complex process, including convection, dispersion, furnace dispersion, adsorption, dissolution, precipitation, hydration and so on [12]. At present, research on heavy metal diffusion mainly includes migration and diffusion experiments but seldom involves mathematical simulation during the diffusion process [13–15]. The diffusion coefficient, dispersion coefficient and retardation factor of heavy metals in soil can be obtained through a mathematical model, which can provide a reference for solving engineering problems of heavy metals in soil. Murray explained the distribution and migration of Pb. The distribution of Pb below the topsoil is related to the content of pollution around the topsoil below the surface. This outcome indicates that Pb will flow and migrate downstream along the seepage zone, even in clay-enriched soils [16]. In kaolin clay, heavy metal migration follows the rule of $Cd^{2+} < Pb^{2+} < Cu^{2+} < Zn^{2+}$ [17]. In this study, we investigated the heavy metal elements in the soil in different areas of the city. The main purpose of the research is to (1) determine the degree of heavy metal pollution; (2) analyze the transmission characteristics and diffusion process of heavy metal pollutants in the soil; (3) establish a pollution source prediction model.

## 2. Data and Modeling Preparation

The establishment of the model is not only to optimize the equation but also to apply it to practice. The data of our model originate from field sampling, and we focus on the migration process of heavy metal pollutants in urban soil. Taking the urban soil of Ningde city, a coastal city in China, as an example, we measured the contents of the surface soil samples by the grid method to provide more research cases for the heavy metal pollution of urban soil and proposed an operable modeling method. The basic research design of this paper is shown in Figure 1.

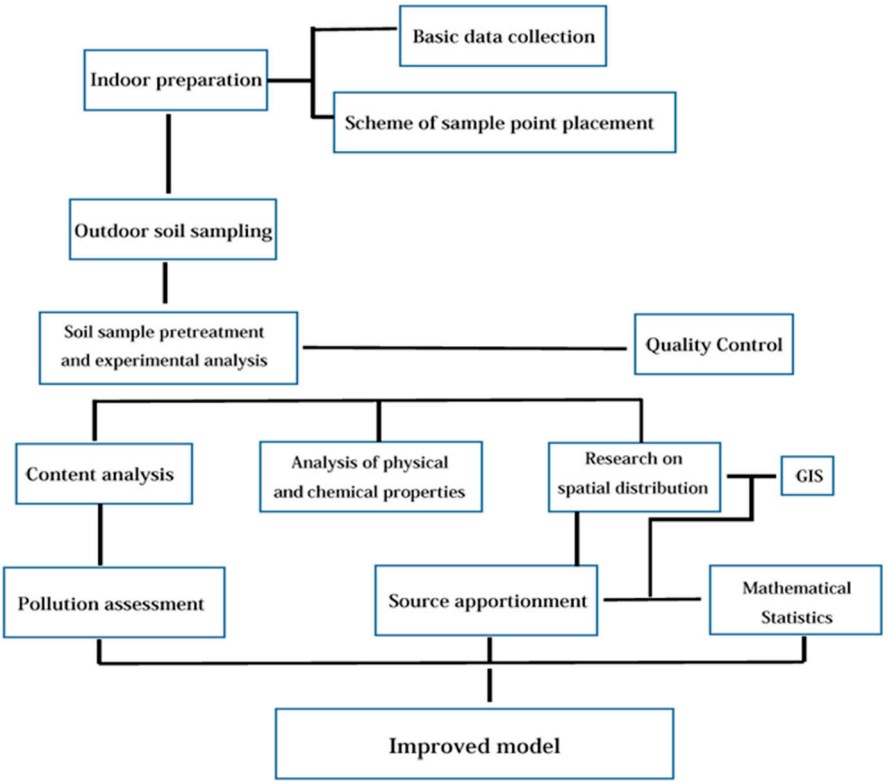

**Figure 1.** Research level and structure.

**Sample collection**

The 1:25,000 Ningde city map was divided into a grid of 500 m × 500 m. From January to December 2019, soil samples from each grid center of Ningde city were collected continuously and located by GPS. For each sample point, five subsample points were arranged according to the "quincunx" shape within 4 square meters, and subsamples of surface soil (0–15 cm) were collected with a stainless steel shovel. Then, the five subsamples were fully mixed to obtain approximately 500 g soil samples.

In the laboratory, the sample was first air dried, and the leaves, bricks and tiles, garbage and other intrusions were removed. Then, the sample was crushed with a wooden stick and passed through a 2 mm nylon screen. Finally, it was mixed fully, approximately 5 g was sampled from multiple points (approximately 30 points) and ground further with an agate mortar to make it pass through the 0.149 mm nylon screen for subsequent experiments.

The urban map of Ningde city is shown in Figure 2. The topographic outline of the urban area is shown in Figure 3. The surface soil was sampled and numbered, and the location of sampling points was recorded with a GPS; in addition, special instruments were utilized for testing and analyzing, a total of 302 samples were obtained. In the actual sampling process, possible contaminated soils such as roads and enterprises were avoided, and GPS was used to accurately locate the sampling points. The sample was dried in the shade, ground with a wooden stick and passed through a 100-mesh nylon sieve to remove gravel, plant roots and other debris, reduced to 100 g using the quarter method, and put it into a plastic bag for testing. A microwave digestion method was used for soil sample determination. A total of 0.200 g of soil sample was weighed, then, 8 mL nitric acid, 4 mL hydrofluoric acid, 4 mL hydrochloric acid were added, and placed in a microwave digestion apparatus. Digestion was performed according to the microwave program. After the end, it was condensed and transferred to a volumetric flask. The heavy metals Pb and Cd in the samples were determined by graphite furnace atomic absorption spectrophotometry, Cr was determined by the flame atomic absorption method, As and Hg were determined by atomic fluorescence spectrometry. The national standard material sample GSS-2 was selected for analysis and quality control. The measurement results showed that the

recovery rates of the five heavy metals were all within 90–110%, which met the international standard reference error.

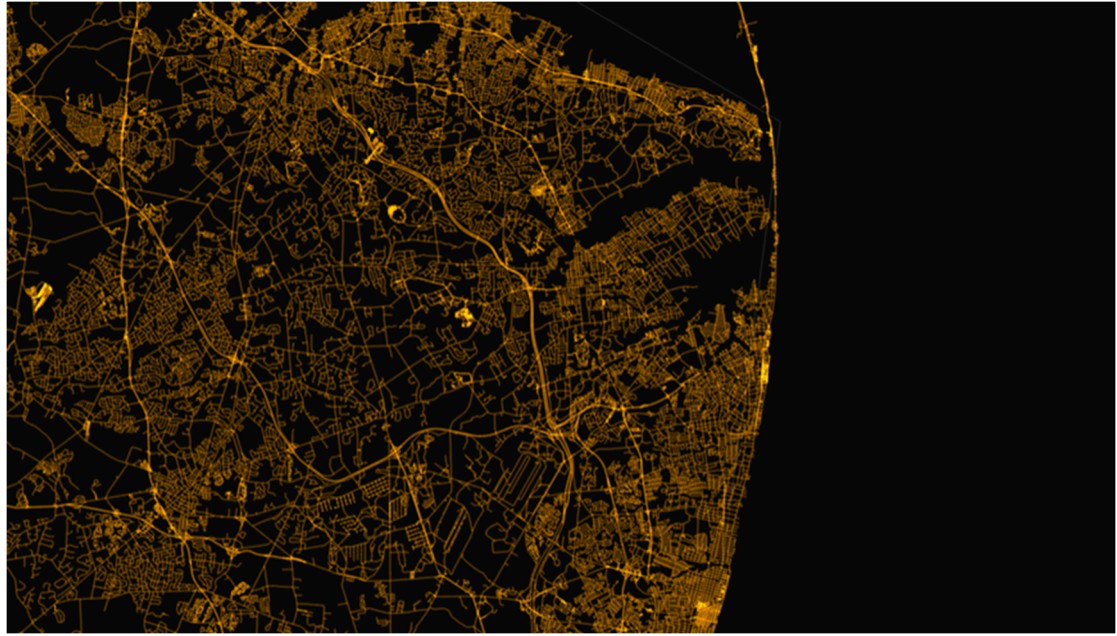

**Figure 2.** The urban map of Ningde city.

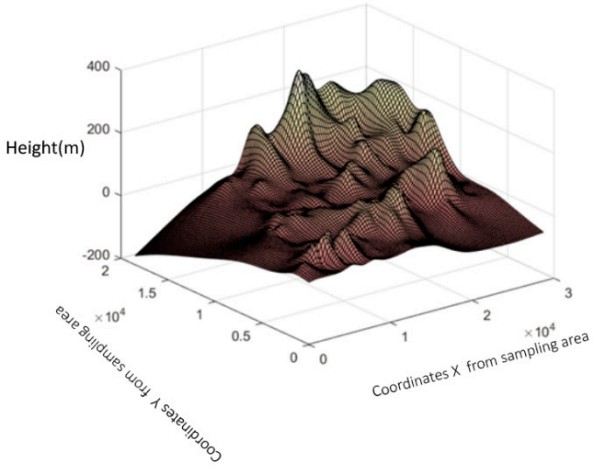

**Figure 3.** Overview of the urban topography.

For the sake of monitoring the urban soil metal element pollution and its diffusion process, determining the geographical location and distribution of As, Cd, Cr, Cu, Hg, Ni, Pb, Zn and other chemical elements in the measured samples is essential. Then, on the basis of spatial distribution, the diffusion process of soil metal element pollution is simulated by a mathematical model. Due to the influence of human activities and climatic factors, the distribution of heavy metal elements in soil satisfies certain randomness and structure. The traditional statistical method does not consider the relationship between the measured parameters and the measured position. It is impossible to perform the optimized interpolation calculation in the region on account of sample data at the different positions in space, and it is impossible to express the geographical distribution of metal elements, such as a time series analysis. The stochastic equation method has its own limitations in dealing with soil heavy metal pollution. Therefore, geostatistics are used in this paper. Because geostatistical methods can effectively describe the spatial variability of soil porous media, kriging interpolation

provides an unbiased optimal estimation method for the spatial prediction of soil properties, so it can sufficiently estimate the distribution of pollutants in soil. Therefore, an increasing number of researchers use geostatistics to carry out kriging spatial interpolation and mapping of soil that has been contaminated by heavy metals [18,19]. Kriging interpolation is an important data processing method in geostatistics. It can be used to detect the spatial autocorrelation structure (or spatial variability structure) of the research object and estimate the value of the simulated variable. Some distribution maps of different metal elements in the soil space were drawn by MATLAB, a commercial mathematics software produced by MathWorks in the United States [20], and the spatial distribution characteristics and variation rules were analyzed.

*Kriging Spatial Interpolation*

The kriging spatial interpolation method is based on the variance function, and its parameter setting and selection of the variance function model have a great influence on the interpolation effect. The calculation of the variogram generally requires that the data conform to the normal distribution, so this paper first tested the normal distribution of the heavy metal concentrations data. There are several methods to check the normal distribution of data, such as frequency distribution histograms and probability plots. A probability plot is a graphic method used to evaluate what kinds of distribution the raw data follow [21]. Here, the probability plot is based on the logarithmic data of the heavy metal concentration in soil, and its normal distribution is tested. Figure 4 is the probability plot of the eight heavy metal elements concentration data.

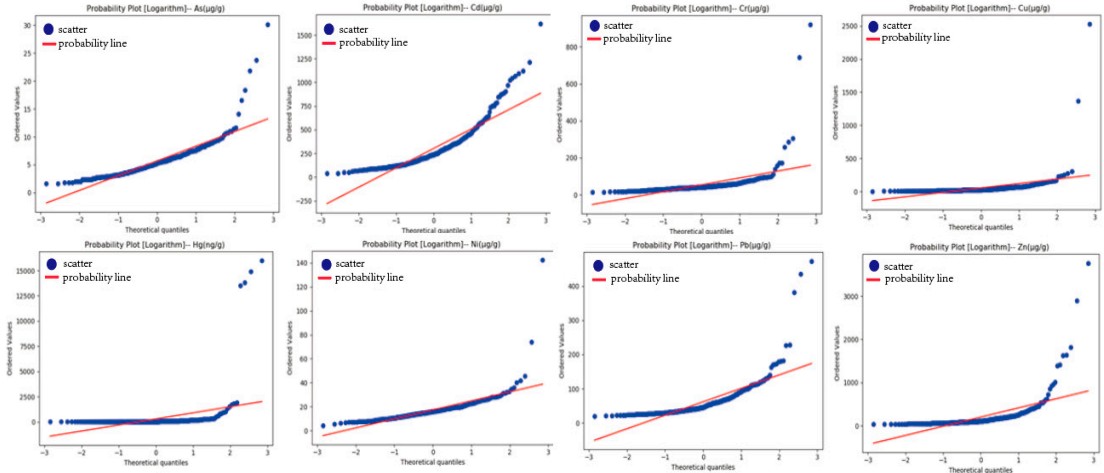

**Figure 4.** Probability plots.

We know that the normal distribution of the eight kinds of heavy metal element concentrations is approximately a straight line, so it can be regarded as following the normal distribution. Therefore, the variogram model can be established to perform the kriging interpolation. When the kriging interpolation is used in the spatial interpolation, all variables are required to conform to the intrinsic hypothesis. The intrinsic hypothesis refers to the stationary hypothesis that is weaker than the second-order stationary hypothesis, requiring that the incremental variance function exists and is stationary (independent of $x$.

$$Z(x) - Z(x + h) \tag{1}$$

$Z(x)$ and $Z(x + h)$ represent the observed values of the two locations within the space region with distances of $h$. That is, when the spatial distance $h$ is small, the correlation between the estimated points and the sample points (elevation points) is high, and the variability is small. In contrast, the correlation between the estimated points and sample points is small, and the variability is large. With the increase in $h$, the variation function $r(h)$ shows a slow increase or no increase. At this time, $r(h)$ is called the

critical variation value. Theoretically, when $h = 0$, $r(h) = 0$, but sometimes there is discontinuity near the origin. The variation function can describe the randomness and structure of the regionalized variables at the same time; in addition, a strict analysis of the regionalized variables in mathematics can be carried out, and it is an effective tool for analyzing the spatial variation rules and spatial structure. The kriging method provides different theoretical variation function models and reflects the spatial variability structure from different perspectives through its structure and various parameters. The commonly used models include the Gaussian model, spherical model and exponential model.

**Gaussian model**

The spatial correlation decreases with increasing distance and tends to zero when the distance tends to infinity.

$$r(h) = C_0 + C_1\left[1 - \exp\left(\frac{-h^2}{a^2}\right)\right] \tag{2}$$

The range of the model is $\sqrt{3}a$. If the kernel variance is small relative to the random variation associated with spatial variation, a more curved Gaussian curve can be selected.

**Spherical model**

The spatial correlation decreases with increasing distance. When the distance is greater than a certain value, the spatial correlation is zero.

$$r(h) = \begin{cases} 0; h = 0 \\ C_0 + C_1\left(\frac{3h}{2a} - \frac{h^3}{2a^3}\right); 0 < h \le a \\ C_0 + C_1; h = a \end{cases} \tag{3}$$

In the above formula, $C_0$ is the nugget constant and represents the amount of random variation. $C_0 + C_1$ is the sill value and represents the structural variance in the spatial variation of variables. $C$ is the arch height, and $a$ is the range; that is, the separation distance corresponding to the curve when it reaches the sill value. The spherical model can be used to fit the semi-variance when there are obvious ranges and the value of the kernel variance is small.

**Exponential model**

Spatial correlation decreases exponentially with increasing distance. When the distance tends to infinity, the correlation tends to zero.

$$r(h) = C_0 + C_1\left[1 - \exp\left(-\frac{h}{a}\right)\right] \tag{4}$$

The range of the model is $3a$. If there is obvious kernel variance without a gradual range, the exponential model can be used to fit it. We know that the concentrations of the eight metal elements are in lognormal distribution, which is suitable for interpolation with the kriging method. The selection of the variogram model compares the advantages and disadvantages of three variogram models, and the root mean square prediction error (RMSPE) is used as the evaluation index of the effect of the variogram model.

$$RMSPE = 2\sqrt{\frac{1}{n_i}\sum_{k=1}^{n_i}\left[Z_i^*(x_k) - Z_i(x_k)\right]^2} \tag{5}$$

$Z_i^*(x_k)$ and $Z_i(x_k)$ represent the detection value and estimated value of $Z$ at the detection point, respectively. When calculating the RMSPE, we randomly reserve a certain number of points from the sampling points as detection points and use other sampling points as the source data of interpolation, and the kriging interpolation is used to determine the estimated numerical numbers of each probe point. Figure 5 is the effect diagram of the three variance functions.

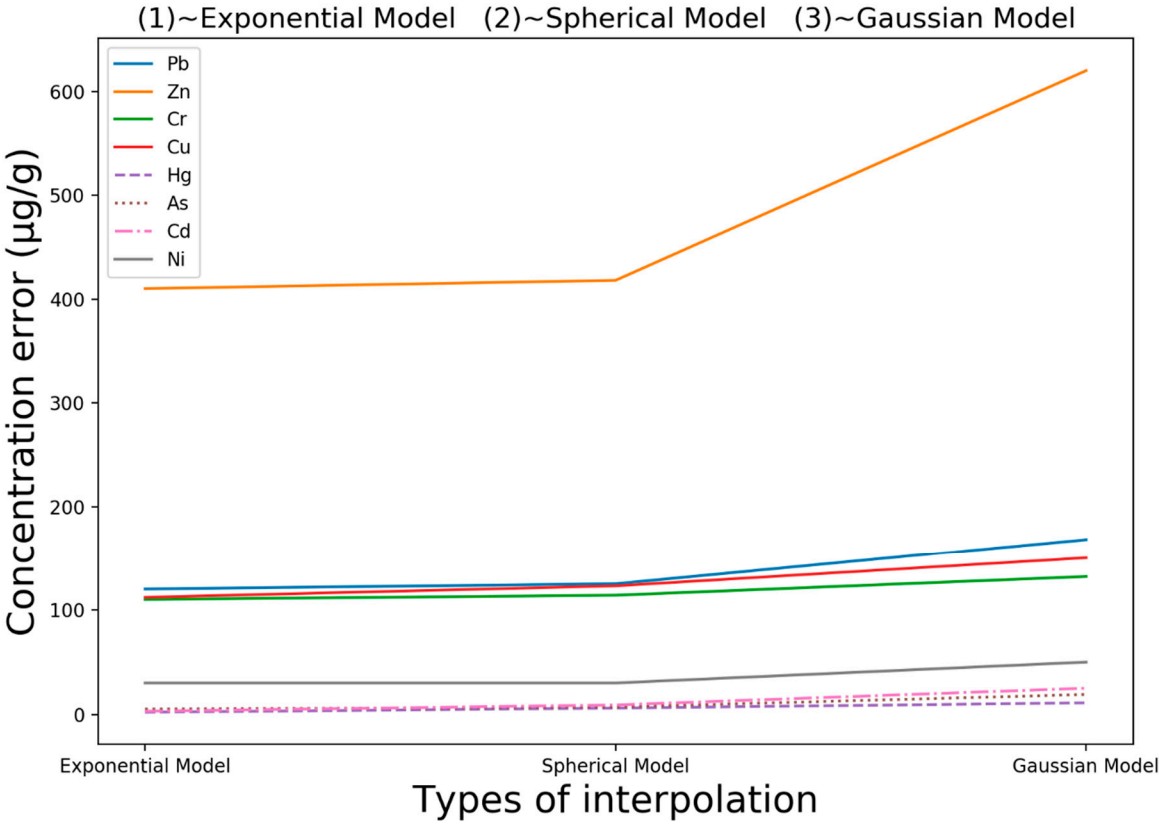

**Figure 5.** The effect diagram of the three variance functions.

In Figure 5, the abscissa axis represents the three variograms: the exponential model, spherical model and Gaussian model, while the vertical axis represents the RMSPE values under the three variograms. As seen from the above figure, the RMSPE of the Gaussian model for the eight heavy metals is obviously higher than that of the first two variograms, so the Gaussian model is not suitable for the kriging interpolation. Compared with the first two models, the RMSPE of the exponential model is larger than that of the spherical model for Cd, indicating that the effect of the spherical model is better than that of the exponential model. Similarly, the interpolation effect of the other seven heavy metals is better with the exponential model than with the spherical model. Therefore, the kriging interpolation based on a spherical model is adopted during the analysis of Cd elements. The kriging interpolation based on an exponential model was adopted to analyze As, Cd, Cr, Cu, Hg, Ni, Pb, Zn After interpolation, Figure 6 is obtained, which is the spatial distribution of the metal concentrations.

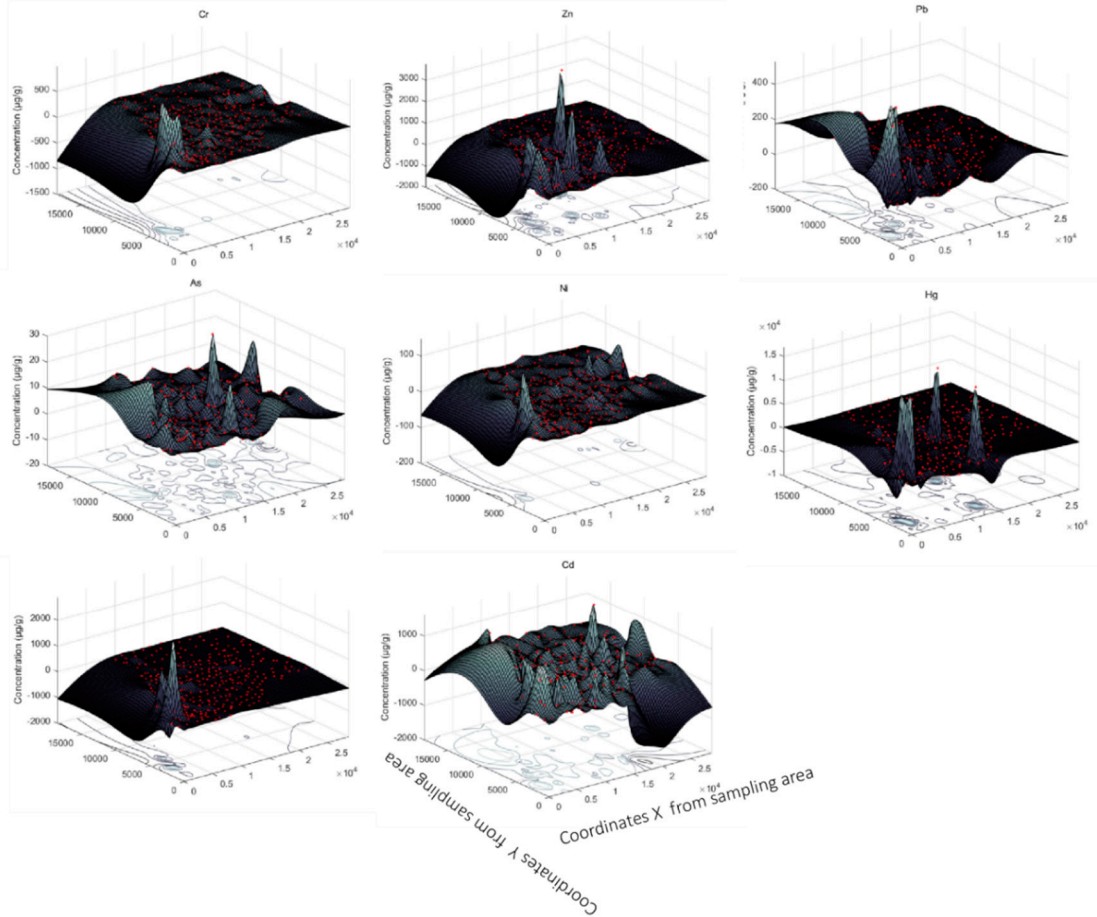

**Figure 6.** The spatial distribution of the metal concentrations after interpolation.

The ratio of the nugget value to sill value is an important index reflecting the degree of spatial heterogeneity of the regionalized variables, also known as the nugget effect. When $\frac{C_0}{C+C_0} < 25\%$, it shows that the spatial variation of variables is mainly structural variation, and the variables have a strong spatial correlation. When $25\% < \frac{C_0}{C+C_0} < 75\%$, it shows that the spatial variability of variables has a moderate degree of correlation; when $\frac{C_0}{C+C_0} > 75\%$, it shows that the spatial variability of variables is mainly random variation, and the spatial correlation of variables is weak.

Table 1 shows that Cd has the strongest spatial correlation, and its $\frac{C_0}{C+C_0}\%$ is much lower than that of the other heavy metals, indicating that Cd pollution mainly comes from natural factors but has a low relevance with human behavior or industrial activities. The spatial correlation of Cr, Hg, Zn, Pb and Ni is moderate. As and Cu have a weak spatial correlation, its $\frac{C_0}{C+C_0}\%$ value is much higher than that of the other heavy metals, indicating that the pollution of As and Cu is mainly related to human behavior or industrial activities.

**Table 1.** Semivariance parameter of the heavy metals.

| Element | Nugget $C_0$ | Structural Variance | Sill Value $C + C_0$ | $\frac{C_0}{C+C_0}$% | Model |
|---------|--------------|---------------------|----------------------|----------------------|-------|
| As | 2.446 | 0.719 | 3.165 | 77.28% | Exponential |
| Cd | 0.002 | 0.009 | 0.011 | 18.18% | Spherical |
| Cr | 3.581 | 2.215 | 5.796 | 61.78% | Exponential |
| Cu | 0.207 | 0.035 | 0.242 | 85.54% | Exponential |
| Hg | 0.257 | 0.192 | 0.449 | 57.24% | Exponential |
| Ni | 1.216 | 2.182 | 3.398 | 35.79% | Exponential |
| Zn | 0.814 | 1.057 | 1.871 | 43.51% | Exponential |
| Pb | 0.916 | 2.490 | 3.406 | 26.89% | Exponential |

After obtaining the spatial map of the interpolation concentration of each metal element and their nugget value, we only need to combine the functional area of Ningde city to obtain the distribution and pollution concentration of each metal element concentration in each functional area of Ningde city. We have drawn the three-dimensional functional zoning map of Ningde city, which is divided into five urban functional areas: living quarters, industrial zones, mountain areas, roads and parklands. Its distribution is shown in Figure 7.

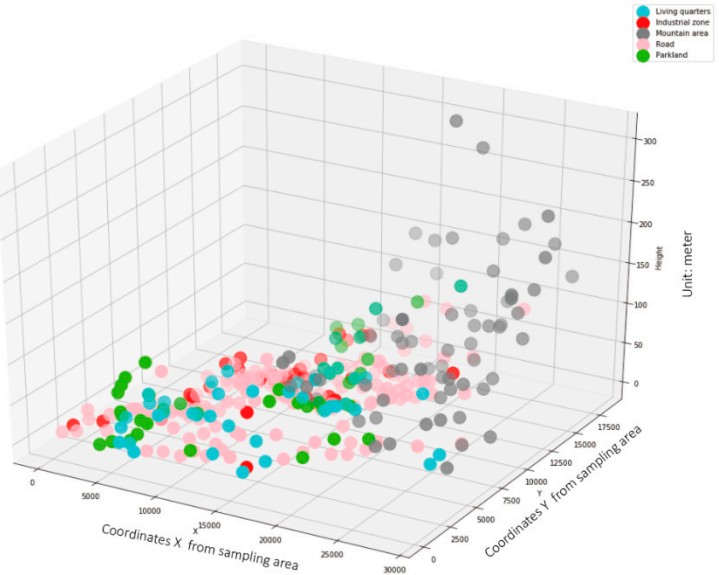

**Figure 7.** Distribution map of the urban functional areas.

The heavy metal contents in soils of the different functional areas in Ningde city are different. The Cu content in soil of the industrial zone is the highest. The order of the Cu content in other functional areas is road > living quarters > parkland > mountain area. The Zn content in the road soil is the highest. The order of Zn content in the other functional areas is industrial zone > living quarters > parkland > mountain area. The highest level of Pb also appears on roads, and the order of the other functional areas is industrial zone > parkland > living quarters. The Ni content is the highest in the industrial zone and the lowest in the mountainous area. Cr content is the highest in the industrial zone, followed by roads, and the difference is not obvious in other areas. As content is the highest in the industrial zone, and there is no significant difference in As in the other functional areas. Cd is distributed in a small number of spots in the industrial zone; in addition, the concentration of Cd in the living quarters and roads is approximately the same and smaller, and the content of Cd in the soil in other areas is very small. The Hg concentration is the highest in industrial zones and roads and the lowest in parklands.

The industrial zone is the area with the highest content of heavy metal pollutants in the five areas. The sources of pollution are mainly from the discharge of production activities, including the discharge of harmful metals from factories, the smelting of metals, the production of dyes [22], etc. For example, mercury (Hg) is produced during the production of mercury compounds in the industrial zone, copper (Cu) is produced during the production of copper products, and other pollutants are produced during smelting and electroplating [22]. In addition, the heavy metal pollution caused by automobiles in roads and the heavy metal pollution brought by automobiles from the industrial zone to the parkland, the living quarters and the mountain area, are also the main causes of heavy metal pollution.

Heavy metal pollution near roads in Ningde city is also serious. The pollutants mainly originate from the exhaust emissions of motor vehicles, as well as the wear and tear of the vehicle body, parts and tires, which release a large amount of harmful dust and gas containing Cu, Zn, Pb and Cd [23]. The rapid increase in vehicle ownership leads to a large traffic flow on urban roads, especially serious congestion during rush hours, the frequent idling of vehicles, the increase in tail gas emissions, and the massive migration of heavy metals to both sides of the road, resulting in line source and even nonpoint source pollution. The heavy metal Pb content of road soil is the highest, although the gasoline in China basically does not contain Pb currently; however, gasoline previously did contain Pb for a long time, which may result in a large amount of Pb remaining in the soil on both sides of the road.

The terrain of Ningde city is dominated by basins. The terrain gradually rises from the southwest to northeast. A small mountain peak appears near the coordinate of 15,000 × 5000, and then it starts to lower. It rises slowly from northwest to east. When it reaches the northeast, the terrain continues to climb to the peak, forming a mountain area in the region. The results show that the concentration of heavy metals is high in the low region and low in the high region. There is a long and narrow river in the south central part of Ningde city, so there are discharge points of pollution sources.

## 3. Pollutant Diffusion Model

The diffusion and migration of metal ion pollutants in the natural environment plays a very important role in pollutant control. Most of the heavy metals in the atmosphere enter the ground by means of natural and rainwater deposition. Simultaneously, due to the rapid expansion of urbanization, large quantities of industrial sewage flooded into the river, causing a large number of heavy metal ions in the urban sewage to flow into the soil with sewage irrigation. The soil pollution near the pollution source and industrial area is more serious, and the degree of soil pollution far away from the pollution source and industrial area is lower. The drainage and rainfall in the polluted area brings heavy metals into the water or directly into the soil, which can indirectly or directly cause soil erosion.

The transport of pollutants inside the ground is affected by various physical, chemical and biological reactions. It is a complex process, including convection, dispersion, diffusion, adsorption, dissolution, precipitation and so on. It is often simplified into the problem of convection–diffusion of pollutants in practice. Currently, research on the mechanism of pollutant migration in soil is more common in soil science and hydraulics. A variety of mathematical models for simulating the migration of pollutants in soil have been established, among which the convective dispersion model is the most widely used [24]. The convective dispersion model only considers the transport of pollutants caused by convection, diffusion and dispersion and the process of adsorption or decomposition of pollutants. The convective–dispersion equation is the main mathematical model describing solute transport and transformation in soil and groundwater.

The basic form of the convective dispersion model:

$$\theta \frac{\partial C}{\partial t} + S\rho \frac{\partial S}{\partial t} = \theta D \frac{\partial^2 C}{\partial x^2} - \theta v \frac{\partial C}{\partial x} - \mu(x)\theta C, S = K_D(x)C \tag{6}$$

In the above formula, $\theta$ is the soil volume water content, $\rho$ is the dry bulk density of the soil, $D$ is the hydrodynamic dispersion coefficient, $v$ is the average pore velocity of the soil water, $C$ is the solute concentration in the soil is the soil solute concentration, which refers to the chemical substance

dissolved in the soil aqueous solution. It is affected by physical, chemical and biological factors. The solute moves with the movement of soil water; under the action of Brownian motion, it also moves from high concentration to low concentration under the action of effective concentration (or activity) gradient. Solute movement is also affected by processes such as adsorption–desorption, crop absorption, precipitation–dissolution, and ion exchange.), $S$ is the solute adsorption concentration (which represents the concentration of heavy metals adsorbed by the soil), $t$ is the time, $x$ is the solute transport distance, $K_D$ is the constant, and $\mu(x)$ is the degradation coefficient (which represents the degradation coefficient.).

The soil properties of the original equation, such as the water content, bulk density and porosity, have a great influence on the simulation results. Among them, bulk density is related to water content, and porosity is related to soil texture. Therefore, the original equation is greatly affected by soil moisture content and soil texture.

## 3.1. Improved Convective Dispersion Model

The traditional one-dimensional convective dispersion model only considers the process of heavy metal pollutants from pollution sources propagating along the one-dimensional horizontal direction of the surface, so the model needs a few parameters and can be applied to most scenarios. With regard to the unidimensional convection diffusion model of pollutants, Shackelford solved the exact solutions of unidimensional convection diffusion of pollutants in semi-infinite clay liners [25]. Acar and Haider provided an exact solution for unidimensional conveyance based on a finite layer thick clay liner [26]. Using Carslaw and Jaeger's solution to the heat conduction problem, Foose obtained the analytical solution of one-dimensional diffusion of pollutants in double-layered soils [27]. It is assumed that the concentration of pollutants remains unchanged, and the content of pollutants in other places is superimposed; in a relatively short period of time, the pollutants will not degrade, and the concentration of pollutants will not decrease. According to the characteristics of pollutant diffusion, the following convective–dispersion equation can be established in the region with only one pollution source.

$$\begin{cases} \frac{\partial W}{\partial t} = H \times \frac{\partial^2 W}{\partial x^2} - B \times \frac{\partial W}{\partial x} \\ W(x,0) = 0; x > 0 \\ W(0,t) = 0; t > 0 \\ W(\infty,t) = 0; t > 0 \end{cases} \tag{7}$$

$W(x,t)$ is a function of $(x,t)$, which represents the net concentration of pollutants at point $x$ (actual concentration minus the background value) at the distance from time $t$ to the pollution source. $H$ and $B$ are unknown equation parameters. $\frac{\partial W}{\partial t} = H \times \frac{\partial^2 W}{\partial x^2} - B \times \frac{\partial W}{\partial x}$ represents the basic convective–dispersion equation. $W(x,0) = 0$ when $x > 0$ is the initial condition. At $t = 0$, the pollutant has not spread, and the nearby site has not been polluted. $W(0,t) = 0$ when $t > 0$ and $W(\infty,t) = 0$ when $t > 0$ represents the boundary condition, that is, the concentration of the pollutant source does not change with time and is not affected by the pollution source at an infinite distance. According to the actual situation, if the pollution in an area is continuous, pollution and the concentration of the pollution source remain unchanged, the variation rate of the pollutant concentration in the vicinity with time is relatively stable. To simplify the calculation, we assume that the variation rate is constant.

$$\frac{\partial W}{\partial t} = v; t > 0 \tag{8}$$

$v$ represents the rate of change in the contaminant concentration over time, which is a small value. For a certain value, the following formula applies:

$$H \times \frac{\partial^2 W}{\partial x^2} - B \times \frac{\partial W}{\partial x} = v \Rightarrow H \times \frac{d^2 W}{dx^2} - B \times \frac{dW}{dx} + v \tag{9}$$

By solving the ordinary differential equation in the above formula, the secular equation is obtained as:

$$H \times \lambda^2 - B \times \lambda - v = 0 (H > 0, B > 0, v > 0)$$
$$\Rightarrow \lambda = \frac{B \pm \sqrt{B^2 + 4H \times v}}{2H} (\lambda_1 > 0, \lambda_2 < 0) \tag{10}$$

The general solution of the ordinary differential equation is $W(x) = C_1 \times e^{\lambda_1 x} + C_2 \times e^{\lambda_2 x}$.

Next, bring $W(\infty, t) = 0$ when $t > 0$ into the general solution to obtain $C_1 = 0$, and bring $W(0, t) = 0$ when $t > 0$ into the general solution to obtain $C_1 = C_2$. In conclusion, the simplified general solution can be obtained as follows:

$$W(x) = C_0 e^{\lambda_2 x} \tag{11}$$

Next, take the logarithm:

$$InW(x) = InC_0 + \lambda_2 x \tag{12}$$

The value of $\lambda_2$ can be obtained by linear fitting, and the diffusion equation of the total metals in soil can be obtained. The above model is not perfect and does not account for the adsorption and decomposition that exist during the convective dispersion process, which is very important; therefore, we have improved it. For heavy metals, the convective dispersion process is often accompanied by a relatively strong adsorption or decomposition process, and crop root absorption and discharge occur. Since the urban soil solution flow is an unsteady flow, we establish an unsteady state migration model.

$$\frac{\partial \theta L}{\partial_{ts}} + \rho \frac{\partial S}{\partial_{ts}} = \frac{\partial}{\partial_z} \left[ \theta D(\theta, q) \frac{\partial L}{\partial_z} - qL \right] - \psi(z, ts) \tag{13}$$

$L$ indicates the metal ion concentrations in liquid, $\theta$ is the ground moisture content, and $\rho$ indicates the dry bulk density of the soil (the bulk density without moisture). $S$ indicates the metal concentrations in the ground solid phase. $D(\theta, q)$ indicates the diffusion coefficient, $q$ is the water flow rate, $ts$ is time (seconds) and $z$ is the soil depth coordinate (meters). $\psi(z, ts)$ is the rate of solute absorption or emission caused by plant roots.

$$\frac{\partial \theta}{\partial_{ts}} = \frac{\partial}{\partial_z} \left[ k(\theta) \cdot \frac{\partial \phi}{\partial_z} \right] + R(z, ts) \tag{14}$$

$k(\theta)$ indicates the transmissivity of the soil. $\phi$ indicates the water potential of the soil. $R(z, ts)$ is the water absorption function of the plant root system.

*3.2. Air Subsidence Model*

3.2.1. Simple Air Subsidence Model

The migration and transmission of heavy metal pollution are not only in the form of underground diffusion but also in the form of atmospheric deposition. The convective dispersion model only considers the one-dimensional horizontal propagation of heavy metal pollutants generated by pollution sources but does not consider the influence of height change and air deposition and diffusion mode on the concentration of metal pollutants among various areas. With respect to the propagation of heavy metal element pollution, air deposition plays an essential role in determining the distribution of metal pollutants. In topographic areas such as canyons and basins, heavy metal pollutants may accumulate, resulting in higher concentrations of pollution than the surrounding areas. Moreover, metal ions basically enter the atmosphere in the shape of aerosols and enter the soil through natural sedimentation and precipitation.

In the air subsidence model, the influence of wind or precipitation should be considered in addition to the influence of agricultural (forestry) activities on the heavy metal concentration

transformation. Subsidence models describe the accumulation of soil solutes caused by wind or precipitation. The variation equation of heavy metal content is as follows:

$$\frac{\Delta M_{ij}}{\Delta t} = I_{Am} + I_{Ar} - Q_L \tag{15}$$

$M_{ij}$ represents the heavy metal concentration in region $i$ and $j$, $t$ represents the time, $I_{Am}$ represents the input flux of heavy metals during air subsidence caused by regional regression, and the unit is $g/(hm^2 \cdot s)$. $I_{Ar}$ represents the input flux of heavy metals caused by agricultural activities, and the unit is $g/(hm^2 \cdot s)$. $Q_L$ is the percolation flux of heavy metals, and the unit is $g/(hm^2 \cdot s)$.

$\frac{\Delta M_{ij}}{\Delta t} = I_{Am} + I_{Ar} - Q_L$ is further transformed to obtain a continuously transformed function.

$$\frac{dM}{dt} = I_{Am} + I_{Ar} - k_c M^m - k_L M^{\frac{1}{n}} \tag{16}$$

$M$ indicates the total metal concentrations, $k_c$ indicates the heavy metal crop absorption coefficient, and $k_L$ represents the heavy metal seepage rate coefficient. $m$ and $n$ are constant.

$$M = \rho z C_t + \theta C_s z$$
$$k_c = \frac{Y_k}{(\rho z)^{b_1 k}} b_0 k \tag{17}$$
$$k_L = q_w \left(\frac{1}{\rho z K_F}\right)^{\frac{1}{n}}$$

$C_t$ indicates the metal concentrations in the soil solution, and $C_s$ indicates the metal concentrations adsorbed by the soil. $Y_k$ represents the crop yield, $b_0 k$ and $b_1 k$ are the regression coefficients. $q_w$ is the Darcy flow rate, and the unit is $L/(m^2 \cdot s)$. $K_F$ is the Freundlich parameter.

### 3.2.2. Improved Air Subsidence Model

The above air subsidence model requires a few parameters to be set, so it can be applied to most scenarios. However, some special scenarios need to consider the impact of the topographic factors and height changes, so the above model needs to be improved. To better determine the major source of the metal element pollutants and analyze the characteristics of metal ions, a Gaussian diffusion equation of the heavy metal pollutants in the atmosphere is established considering the topographic height and air subsidence. The pollution source is selected as the coordinate origin, and the concentration of metal ions around the polluted zone is recorded as $W(x, y, z, 0)$. The concentration of heavy metals at any point in the infinite space from the pollution source $j$ is $W(x_i, y_i, z_i, j_i)$. The flow rate through the unit normal zone varies directly as a concentration gradient within a unit distance.

$$\vec{q} = -\delta_1 \cdot grad W \tag{18}$$

$\delta_i (i = x, y, z)$ is the diffusion coefficient, and $grad$ is the gradient. A negative sign indicates a spread from a high concentration to a low concentration. The space domain is $\Omega$, and its volume is $V$. The surface surrounding $\Omega$ is $S_1$, and $S_1$ is a regular sphere. The outer normal vector of $S_1$ is $\overline{n} = \left(\frac{-x}{z}, \frac{-y}{z}, 1\right)$. The flow through $\Omega$ is:

$$T_1 = \int_j^{j+\Delta j} \int \int_{s_1} \vec{q} \cdot \vec{n} \, d\sigma dj \tag{19}$$

The increment of heavy metal elements in $\Omega$ is:

$$T_2 = \int \int \int_V [W(x, y, z, j + \Delta j) - W(x, y, z, j)] dV \tag{20}$$

The total amount of heavy metals released from the pollution sources is as follows:

$$T_0 = \int_j^{j+\Delta j} \int\int\int_\Omega p_0 dv dt \tag{21}$$

According to the principle of conservation of mass and the principle of continuity of gas leakage, the sum of the amount of the heavy metal elements diffused outward through the selected surface $S_1$ in a unit distance, and the increment of heavy metal elements in a surface is equal to the amount of heavy metal elements released outward by pollution sources in a unit distance.

$$T_0 = T_1 + T_2 \tag{22}$$

There are Gaussian formulas for surface integrals, where *div* is a divergence mark.

$$\int\int_{S_1} \vec{q} \cdot \vec{n} d\sigma = \int\int\int_V div\vec{q} dV \tag{23}$$

$$\int\int\int_V \left[\frac{\partial W}{\partial j}\right] dV + \int\int\int_V div\vec{q} dV = \int\int\int_\Omega p_0 dV \tag{24}$$

From the mean value theorem of integrals:

$$\frac{\partial W}{\partial j} = \delta_i div(gradW) = \delta_i\left(\frac{\partial^2 W}{\partial x^2} + \delta_y \cdot \frac{\partial^2 W}{\partial y^2} + \delta_z \cdot \frac{\partial^2 W}{\partial z^2}\right)$$
$$j > 0, -\infty < x, y, z < \infty \tag{25}$$

The initial condition is the source of pollution at the origin of the coordinates.

$$W(x, y, z, 0) = T_0\delta(x, y, z) \tag{26}$$

$T_0$ represents the total amount of heavy metal elements released, and $\delta(x, y, z)$ is the diffusion coefficient of the unit strength.

$$W(x, y, z, j) = \frac{T_0}{(4\pi k\delta_i j)^{\frac{3}{2}}} e^{\frac{x^2+y^2+z^2}{4kS_i j}}, i = (x, y, z) \tag{27}$$

where $k$ is the speed of propagation in time $t$. The results show that for any time $t$, the contaminant concentration $W$ is spherical $x^2 + y^2 + z^2 = R^2$. As the spherical radius $R$ increases, when $R \to \infty$ or $t \to \infty$, $W \to \infty$. R can be solved by substituting $x^2 + y^2 + z^2 = R^2$.

$$R^2 = (-4)kt \cdot In\left(\left(\frac{W}{T}\right) \cdot 4\pi kt^{\frac{3}{2}}\right) \tag{28}$$

$R$ is the radius of the range affected by the natural diffusion under a given diffusion coefficient $k$, influence time $t$, pollutant concentration $W$, and total pollutant $T$, that is, the range affected by the natural diffusion without considering wind and rainfall. Since the wind direction and wind speed are extremely random and cannot accurately quantify the wind direction and wind speed, the model chooses to simplify the assumption, ignoring the influence of wind direction and wind speed.

In general, the improved model belongs to the vertical migration model. Currently, there are models for horizontal migration in academia, such as the SWAT model (Soil and Water Assessment Tool). The SWAT model is a watershed scale model developed by the U.S. Department of Agriculture. It is used to simulate the change in pollutants in a complex watershed with a variety of soil, land use and management conditions and to predict the impact of long-term land management measures on water, sediment and agricultural pollutants. The SWAT model shows a high precision in data

processing. The mature soil loss models including the revised universal soil loss equation (RUSLE) in the model are closely linked with GIS software, which can fully utilize the data processing ability of ArcGIS, a mapping and analytics platform, using contextual tools to visualize and analyze data. We carried out a sensitivity analysis on the model mentioned in the article, screened out the results that have a high impact on the numerical results of the model, and compared the vertical migration model with the horizontal migration model. The comparison table is shown in Table 2.

**Table 2.** Sensitivity parameters of the different models.

| Migration Mode | Simulation | | Sensitivity Factors in the Model | | | | |
|---|---|---|---|---|---|---|---|
| | | | Heavy Metal Properties | | Soil Physical and Chemical Properties | Vegetation Properties | |
| | | Type | Form | | Organic Matter, pH Value, Texture and Moisture Content | Cumulative Effect | Climatic Factors |
| | | | Degree of Influence | Degree of Influence | Degree of Influence | | |
| Vertical migration | Integral model | Convective dispersion model | / | +++ | ++ | / | / |
| | | Air subsidence model | + | / | / | ++ | +++ |
| | | Improved model | + | / | ++ | ++ | ++ |
| Horizontal migration | SWAT model | ++ | ++++ | +++ | +++ | +++ |

The influence of heavy metal species and climate factors on the model is universal, and the influence mechanism is basically the same. They are not listed in this table. The "+" in the table indicates the degree of influence.

### 3.3. Propagation Characteristics of Pollutants

After taking into account factors such as adsorption or decomposition during the process of convective dispersion, absorption and discharge of the crop root system and topographic factors and height changes during the process of air settlement, the improved one-dimensional convective dispersion model and the improved air subsidence model were established. Then, the parameters and concentration distribution formulas of the original equation were obtained. According to the formula obtained, we set different diffusion times to determine changes in the metal elements in urban ground to obtain the propagation features of the metal ions [28]. Based on the formulas and actual conditions obtained above, we set the parameters $B = 12$ and $H = 4$. The general solution $C(x) = C_0 e^{\lambda_2 x}$ was solved by programming. Figure 8 is the propagation characteristics chart of the heavy metal pollutants.

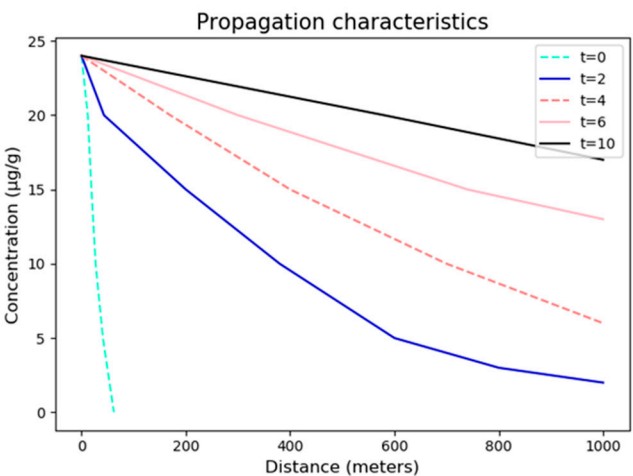

**Figure 8.** Propagation characteristics chart of the heavy metal pollutants.

As shown in Figure 8, with increasing time *t*, the distance of pollutant propagation increases gradually, and the concentration of pollutants in the same place increases gradually. This outcome shows that under the continuous pollution mode, the concentration of pollutants in the region becomes cumulative.

## 4. Discussion

### 4.1. Prediction of Pollution Source Location

After obtaining the propagation features of metal ions, we need to speculate the locations of the pollutant sources. First, we found the abnormal maximum value region of each heavy metal element, and used the local search method to fit the concentration distribution formula with the actual data Next, the points whose goodness of fit values in the four directions are all fitted are selected as the pollution sources in the region. The final result of the determined pollution source is shown in Figure 9.

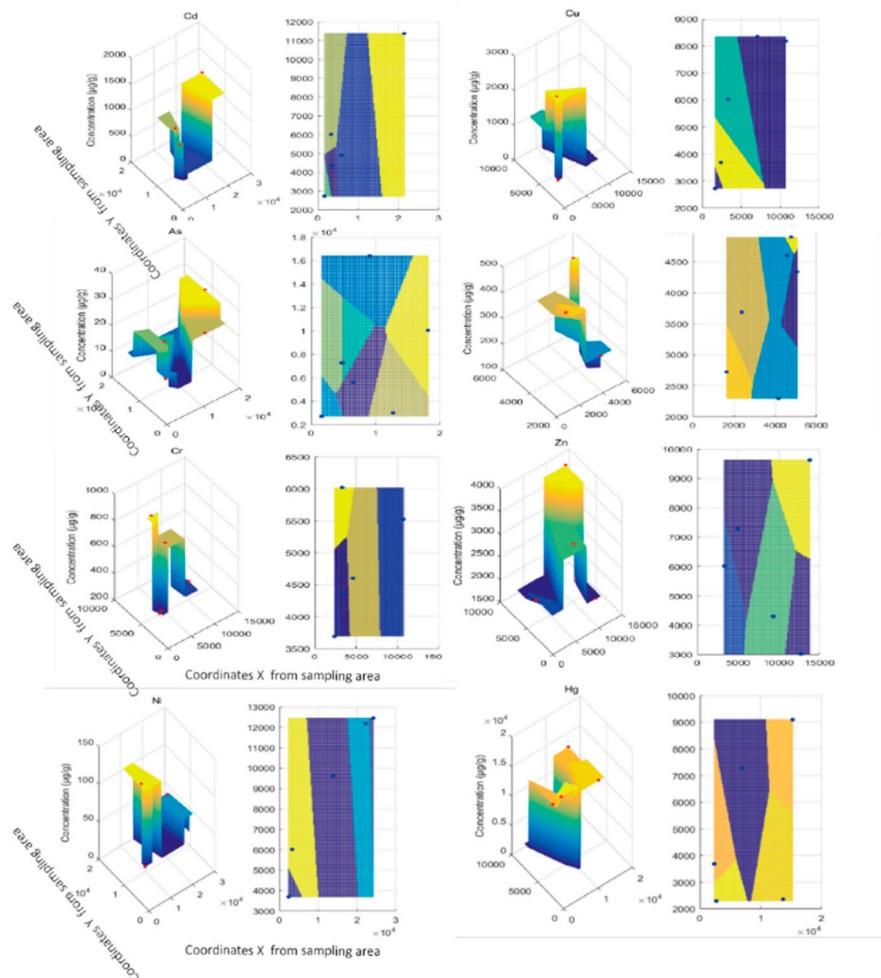

**Figure 9.** Stereo and plane views of the heavy metal pollution sources.

The spatial distribution of the major metal pollution sources is shown in Figure 9, in which the figure on the left side is a stereogram and the right side is a plane. The different colors represent the concentration of pollutants. The yellower, the higher the concentration of pollutants. The bluer the color in the figure, the lower the concentration of pollutants. It can be seen from the results that the number of heavy metal pollution sources is usually more than one and may be far apart. The coordinates presented in Figure 10 can be used to determine the location of the pollution source in reality.

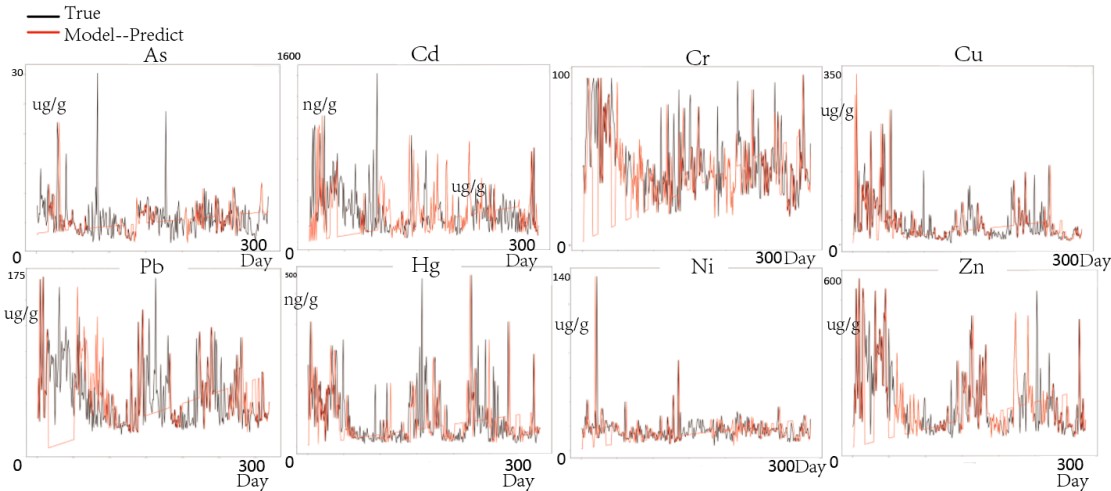

**Figure 10.** The effect of the improved model.

*4.2. Prediction Effect of the Improved Model*

We have derived the propagation characteristics of the improved model, calculated the unknown parameters in the propagation equation, and attempted to preliminarily locate the source of pollutants. The above steps are the extension and application of the improved model, but we need to ensure that the fitting effect of the model is better and that the control error is as small as possible. Therefore, determining how to measure the improvement of the above model and how to ensure the improvement effect of the model are crucial.

We fitted As, Cd, Cr, Cu, Hg, Ni, Pb, Zn and other elements and compared the simulated value with the real value, as shown in Figure 10. Here, the red line is the predicted simulation value, and the black line is the real value. We used the improved model to predict a total of 365 days in 2019. The abscissa represents the time, and the ordinate represents the concentration of each heavy metal pollutant.

From the above images, we can see that the fitting effect of the model is not good at the beginning, and there are many vertical fluctuations during the middle process. This sudden fluctuation is caused by the parameters set by the improved model. With the passage of time, the prediction of the improved model for the concentration of heavy metal pollution in the soil is increasingly accurate, and the difference with the real value is gradually reduced. We introduce the concept of mean-square error (MSE) in statistics to describe the numerical difference between the predicted value and the real value of the model [29]. Here, we give the MSE values of each heavy metal element: As—12.14, Cd—361.18, Cr—563.15, Cu—314.07, Hg—541.415, Ni—192.85, Pb—751.12, and Zn—844.69.

## 5. Conclusions

The diffusion and migration of heavy metal pollutants in soil and water media plays a very important role in pollutant control. Most of the heavy metals in the atmosphere enter into the soil through natural and rain deposition. At the same time, due to the rapid development of urban industrialization, a large amount of industrial wastewater flows into the river, which makes a large number of heavy metal pollutants in urban sewage flow into the soil with sewage irrigation. The migration of pollutants in soil is affected by various physical, chemical and biological reactions, which is a complex process, including convection, dispersion, diffusion, adsorption, dissolution, precipitation and so on.

This paper focuses on the problem of heavy metal pollution in urban surface soils, answers the question of how to use mathematical models to process massive amounts of data, analyzes the transmission characteristics and diffusion process of heavy metal pollutants, and establishes a pollution source estimation model. This article first uses the Kriging interpolation method in geostatistics to

simulate the distribution of chemical elements such as As, Cd, Cr, Cu, Hg, Ni, Pb, Zn. For each heavy metal element, the exponential, spherical and Gaussian variation models were established, respectively, and the smallest RMSE predicted value among the three functions was selected as the variation function. Then, Matlab was used to draw the three-dimensional spatial distribution map of each element, and through the ratio of the nugget value to the abutment value, it can be concluded that the pollution of Cd mainly comes from natural factors, but has a small connection with human behavior or industrial activities; the spatial correlation of Cr, Hg, Zn, Pb, and Ni is medium. The pollution of As and Cu is mainly related to human behavior or industrial activities. Then, a one-dimensional convection dispersion equation was established for heavy metal elements in soil, and the parameter values and concentration distribution formula of the original equation were obtained. Different times of diffusion were used to analyze the content changes of heavy metal elements in different regions, and the transmission characteristics of heavy metal pollutants were obtained. Then, the improved one-dimensional convection dispersion model and the improved air deposition model were established, respectively. Compared with the original model, more factors were considered, such as the adsorption or decomposition factors in the process of convection dispersion, the absorption and discharge factors of crop roots, and the terrain factors and height changes in the process of air deposition. The abnormal maximum area of each heavy metal element was searched, and the concentration distribution formula was fitted with the actual data by using the local search method, and the points with the goodness of fit in four directions were selected as the pollution sources in the region. Finally, the main pollution sources of heavy metals were obtained.

Compared with the original model, our model's advantage is that because the transport of pollutants in the soil is a very complex process, it is a process of integrating physical, chemical and biological actions, including convection and diffusion, adsorption and desorption, attenuation and accumulation. In the process, we not only consider the effects of diffusion adsorption and microbial degradation in the model, but also establish a mathematical model of convection and diffusion of pollutants in saturated and unsaturated soils based on the principle of conservation of mass.

The disadvantages of this paper are that (1) the sample point is small, and the data of only one city have not been verified on a larger scale. (2) The lack of data, such as wind direction and speed, is also an important factor restricting the improvement of the model. (3) The chemical nature of pollutants and changes in the chemical form of heavy metals are not considered. The chemical forms and proportions of heavy metals in the environment are closely related. The form of heavy metals can determine their toxicity, bioavailability and migration ability in environmental media [30–32]. However, because the mechanism conversion between different forms is too complicated, we generally assume that it is constant during the mathematical model process, which is also an inherent drawback of the mathematical model.

**Author Contributions:** Conceptualization, Z.L.; methodology, Z.L.; validation, J.H., W.L.; formal analysis, W.L.; resources, W.L.; data curation, J.H.; writing—original draft preparation, J.H., W.L.; writing—review and editing, Z.L.; visualization, W.L.; supervision, Z.L.; project administration, Z.L.; funding acquisition, Z.L. All authors have read and agreed to the published version of the manuscript.

**Funding:** This research was supported by the National Natural Science Foundation of China (Grant No. 71974057).

**Conflicts of Interest:** The authors declare that there is no conflict of interests regarding the publication of this article.

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
