# Peer review of "Monitoring, Diffusion and Source Speculation Model of Urban Soil Pollution"

_processes, doi:10.3390/pr8111339_

Round 1

Reviewer 1 Report

Dear authors,

I find your model a great tool to solve soil pollution by heavy metals. However, I feel there are some gaps in your article. Overall I think your article needs some chapter re-structuring (there is a lot of repeated and misplaced information), both the figures and models are poorly described (the reader has a lot of issues in drawing any conclusions) and some points of the methodology are not clear either. I would also advise you to add some references to support your findings. Finally, you do not clearly state the addition your paper makes to the field, or when you do it, this is not backed by your results and conclusion (you probably have the results to back it, but you don´t show it throughout the article). Please find a point-by-point review in the doc attached.

Best regards

Author Response

To: Processes Reviewer 1

Re: Response to reviewer

Dear Reviewer 1,

Thank you for your suggestions on our manuscript.

Thanks for your recognition of our article model. Based on your suggestions, we re-adjusted the article structure, language and conclusions. Please see below for specific operations, and we have responded to each of your suggestions.

Best regards,

<Wanchun Lu> et al.

Reviewer 2 Report

This manuscript suggests a mathematical model for analysis of convective diffusion process of heavy metal pollutants. The model can be used to identify the source of the pollution. The authors focus on the migration of As, Cd, Cr, Cu, Hg, Ni, Pb, Zn in urban soil of Ningde city (China). The results presented can be interested for researchers involved in the environmental investigation and remediation processes.
The paper also can be useful to the scientists interested in modeling of the processes of diffusion and convective transfer. However, before that major revision of the manuscript and extensive editing of English language and style are required. The authors should correct and explain some statements.

  • At the moment the introduction is vague, includes non-relevant references and it is not focused on the main problem. The introduction should be shortened and improved.
  • Line 88: «Abundant trimethyl» - What chemical compound do the authors mean?
  • Lines 90-91: What chemical form of lead (and other metals), what the industrial processes and polar microparticles attached 91 to the air do the authors mean?
  • Careful editing of text is required, for example, lines 114, 116, 119-120, 133, 156 ….
  • Extensive editing of figures and tables is required: the axis labels and the units of measure are absent (table 1, fig.4, 7, 8, 10, 11); the font is very small.
  • 161-163: I would suggest authors to clarify what «special instruments» for determining of As, Cd, Cr, Cu, Hg, Ni, Pb, Zn concentrations were used. What technique of elemental analysis of the samples was used?
  • Lines 298-305: The authors said that “mercury (Hg) is produced during the production of mercury compounds in the industrial zone”. What kind of mercury production facilities operate in the industrial area? Do they really work without any waste treatment facilities? What kind of “heavy metal pollution caused by automobiles»?
  • Lines 346-347: What “the solute concentration in the soil» and «the solute adsorption concentration», «the degradation coefficient” do the authors mean? I think, the authors should give formulas for calculating these values.
  • Line 397: “water potential of the soil» – how to determine this value?
  • Table 2 is incomprehensible. Can authors change it for better understanding?
  • Another important question is how does the chemical state of metals change? The authors absolutely do not take into account the chemical properties of pollutants. Please explain. The authors need to provide some literature or supporting evidence if available.

Author Response

To: Processes Reviewer 2

Re: Response to reviewer

Dear Reviewer 2,

Thank you for your suggestions on our manuscript.

Thanks for your recognition of our article model. Based on your suggestions, we re-adjusted the article structure, language and conclusions. Please see below for specific operations, and we have responded to each of your suggestions.

Best regards,

<Wanchun Lu> et al.

Round 2

Reviewer 2 Report

Thanks to the efforts of the authors, the language, structure and presentation of the manuscript have been significantly improved. I would like to draw your attention to the 'Conclusion'. As for me, the authors should improve clarity of the 'Conclusion' (Lines 620-630).

Author Response

Dear Editor,

Thank you for your suggestions on our manuscript.

According to your suggestions, we have done two aspects of work. (1) Revise and improve the conclusion of the article, simplify and clarify the practice and contribution of the article. (2) Proofread the full text of the language and improve the grammatical errors of the article.

Best regards,

<Wanchun Lu> et al.
